# Structural Designs of Transparent Polyimide Films with Low Dielectric Properties and Low Water Absorption: A Review

**DOI:** 10.3390/nano13142090

**Published:** 2023-07-17

**Authors:** Sivagangi Reddy Nagella, Chang-Sik Ha

**Affiliations:** Department of Polymer Science and Engineering, School of Chemical Engineering, Pusan National University, Busan 46241, Republic of Korea; sivagrnagella@gmail.com

**Keywords:** polyimides, dielectric properties, low D_k_, low D_f_, low water absorption, 5G/6G communication

## Abstract

The rapid development of communication networks (5G and 6G) that rely on high-speed devices requiring fast and high-quality intra- and inter-terminal signal transmission media has led to a steady increase in the need for high-performance, low-dielectric-constant (D_k_) (<2.5) materials. Consequently, low-dielectric polymeric materials, particularly polyimides (PIs), are very attractive materials that are capable of meeting the requirements of high-performance terminal devices that transmit broadband high-frequency signals. However, such a PI needs to be properly designed with appropriate properties, including a low D_k_, low dielectric loss (D_f_), and low water absorptivity. PI materials are broadly used in various fields owing to their superior property/processibility combinations. This review summarizes the structural designs of PIs with low D_k_ and D_f_ values, low water-absorbing capacity, and high optical transparency intended for communication applications. Furthermore, we characterize structure–property relationships for various PI types and finally propose structural modifications required to obtain useful values of the abovementioned parameters.

## 1. Introduction

Increasing demand for the Internet of Things (IoT) requires the development of communication technologies, such as 6G, and the further advancement of 5G communication capabilities [1]. With 5G/6G wireless technologies, connections between base stations and mobile phones/intelligent machines are more reliable, enabling faster inter-device communication [2]. Therefore, such technologies provide instantaneous and uninterrupted communication between electronic devices, leading to a broad range of new applications, such as autonomous transportation vehicles, digitized logistics, augmented reality, smart retail and smart home applications, remote diagnosis, robotic surgery, patient care in healthcare systems, and precision and sustainable agriculture, thereby enabling a smart and fully connected world [2,3].

Compared to wired communication, wireless communication, which transmits data through space, is somewhat constrained owing to limited transmission rates and sizes. Mobile communication involves transmitting information using electromagnetic waves that propagate to the destination via multiple pathways, which include scattering, reflection, diffraction, and other phenomena that reduce the signal-to-noise (S/N) ratio and affect signal quality. Accordingly, high-frequency signals with higher bandwidths are required to solve this problem and deliver maximum transmission rates. Thus, 5G and 6G technologies that use larger-bandwidth, high-frequency electromagnetic waves are capable of solving this problem. The transmission rate, S/N ratio, and bandwidth are related by the Shannon formula [4]:(1)$$C=W\;{log}_{2}(1+\frac{S}{N})$$
where C is the maximum transmission rate, W is the spectral width, and N and S are noise and signal power. 

Due to recent technical advancements, electronic devices have become increasingly smaller [5]. These miniatured electronic gadgets need to be highly integrated and respond quickly; consequently, materials with low dielectric constants (D_k_ values) and dielectric losses (D_f_ values) are required to meet the demands of the microelectronics industry for the use and manufacture of 5G/6G communication devices [5,6,7]. Furthermore, their high heat dissipation and low dielectric constants are suitable for fast-signal-transmission applications. The heat generated in a communication device is mainly attributable to integrated circuits (ICs), light-emitting diodes (LEDs), and radio-frequency power devices [8]. Consequently, integrated materials are required to withstand and dissipate heat well; otherwise, residual thermal stress can destroy the internal structure of the electronic device and irreversibly damage it. In addition, the signal transmission speed and quality (without signal loss) greatly depend on the optimal internal transmission time (signal delay time) between highly integrated electronic components [9]. The following relationships exist between the signal delay time (t_d_) and dielectric constant and between the signal loss (L) and dielectric loss tangent (tan δ):(2)$$t_{d}=\frac{l\times \surd \varepsilon_{r}}{C}$$
(3)L=a×f×tanδ 
where ε_r_ is the relative dielectric constant (permittivity), l is the transmission distance, a is a constant, and f is the frequency of the electromagnetic signal. Here, the signal gain or signal transmission quality of the receiver mainly depends on the dielectric material because the resonance frequency of the electromagnetic signal is a function of its thickness and dielectric constant. Therefore, the major disadvantage of dielectric material is a narrow bandwidth, which affects performance, because the desired substrate should maintain signal gain throughout the intended bandwidth (4G to 6G). In this sense, reducing the dielectric constant and loss tangent of the dielectric material improves the quality of the transmitted signal by decreasing its parasitic capacitance and signal time delay. Hence, the characteristics and functionalities of a dielectric material are very important in terms of device quality and performance [10]. 

## 2. Materials with Low D_k_ and Low D_f_ for 5G/6G Applications

The development of 5G/6G communication technologies has led to significant efforts directed toward increasing the network speed and the number of simultaneous connections [11,12,13]. Furthermore, millimeter-wavelength radio waves suffer from high attenuation, leading to significant signal-quality losses [14,15], which can be overcome by installing a large number of beamforming antennas (an array of antennas that controls the direction of the wavefront), but this requires high-quality insulating materials with low D_k_, low D_f_, poor water absorptivity, and good adhesiveness to maintain high-level substrate wiring (referred to as “flexible copper-clad laminates”). A few materials are suitable for 5G/6G applications, including polytetrafluoroethylenes (PTFEs), some ceramics, and liquid-crystalline polymers (LCPs) [16,17,18]. However, these materials suffer from their own drawbacks: for instance, ceramic materials require high-energy procedures (longer, high-temperature processing). Even though the cold sintering process consumes less energy, it decreases the density and degrades the microstructure of the ceramic [17]. PTFEs exhibit good heat and corrosion resistance, as well as optimal dielectric properties; however, they suffer from high costs and environmental regulatory restrictions [16]. Consequently, LCPs have mainly been used as insulating materials to date. LCPs have recently gained a lot of interest as packaging and circuit materials for 5G/6G communication devices, as well as IOT applications, due to their high efficiency rates and low loss properties in broadband transmission [19]. LCPs are preferred over FR-4 or G10 because of their expanded working temperature range (operational range from −60 to +260 °C) and superior dielectric properties. They are also compatible with copper, gold, titanium, and other extensively used metals and semiconductors. However, they suffer from several severe problems, like the tearing problem during the manufacturing process, the bad adhesion and poor reliability of the molded plates in the early stage of injection (up to 1/3 volume), and low dielectric stability above 100 °C. But polyimides have low coefficients of thermal expansion (~20 ppm/°C) and high tensile strength, and thus, they can withstand higher temperatures (>400 °C). Further, the synthesis and fabrication of PIs are simple and cost-effective [20,21]. Nowadays, therefore, polyimides (PIs) are replacing LCPs because of their unique advantages. 

## 3. PIs for 5G/6G Applications

### 3.1. General Properties of PIs in Terms of 5G/6G Applications

PIs have been used as high-performance polymeric materials in various applications owing to their high glass transition temperatures (T_g_s), high thermal stability, and good mechanical and optical properties; such applications include engineering plastics, insulating coatings for aerospace applications, membrane-separation films, and various optical and electronic devices [22,23,24,25,26]. Furthermore, thermally stable PIs facilitate a higher upper operating temperature because they provide good dimensional stability above their T_g_ values [27]. Additionally, fluorinated aromatic or semi-aromatic PIs have the additional advantages of optical transparency and lower D_k_ and D_f_ values. Research into fluorinated PIs with aromatic or semi-aromatic backbones has recently become important for delivering high-performance 5G/6G communication devices [28]. 

Aromatic polyimides (APIs) form an important family of heteroaromatic polymers for use in advanced electronic, microelectronic, and modern 5G and 6G communication devices owing to their advantageous properties, which include high thermal stability, good mechanical strength, good insulating properties, and low D_k_ values [29]. Furthermore, they can be processed into various materials, such as films, fibers, composites, foams, porous membranes, coatings, and engineering materials.

APIs are highly thermo-oxidatively stable, non-flammable, and heat-resistant; their high T_g_s (up to 400 °C for amorphous resins) and very high melting points facilitate their use in the −269 to +400 °C temperature range [30]. The high heat resistance of a PI is mainly attributable to its high T_g_ and amorphous nature. Furthermore, PIs have excellent mechanical properties, including high moduli and tensile strength, which facilitate their use in the wide −270 to +300 °C temperature range [31]. Even though PIs have suitable properties for high-speed communications, they have some drawbacks, including poor solubility in organic solvents, infusibility, and strong coloration due to charge-transfer-complex (CTC) formation [32,33,34].

Research into APIs for optoelectronic and telecommunication applications has flourished in recent years. Although comprehensive reviews on the subject are uncommon, several review papers have been published in different areas. For instance, Tapaswi and Ha [35] and Ni et al. [10] independently published review papers on colorless and transparent PIs, whereas Li et al. reviewed low-dielectric PI films [6], and Hasegawa reviewed transparent PIs with ultralow linear coefficients of thermal expansion (CTEs) [36]. The current review article summarizes recent research publications and systematically examines relationships between structure and properties from the perspectives of dielectric and water absorption properties and optical transmittance, with methodologies required to produce high-performance PI materials for 5G and 6G communication devices reviewed. Furthermore, this review briefly looks toward future research directions for optically transparent PIs with low D_k_ and D_f_ values that also absorb water poorly.

### 3.2. Properties of PIs That Need to Be Considered for 5G/6G Applications

#### 3.2.1. Dielectric Properties

The dielectric performance of a material is measured in terms of electrostatic energy storage and loss in the presence of an electric field and is expressed in terms of its dielectric constant and dielectric loss [37,38,39], which are important parameters used to evaluate the electrical insulating performance of a material. The dielectric properties of any dielectric material are well known to mainly depend on internal factors, including various types of polarization phenomena, and external factors, such as frequency, temperature, and humidity [40]. Electric dipoles are created in an external electric field as electrons shift with respect to the atomic nuclei as centroid electron clouds depart from their centroid nuclei. Consequently, a secondary electric field (that satisfies the resultant field and polarization vector) is produced when a dielectric material is exposed to an electric field due to induced dipoles: (4)$$\vec{P}=\varepsilon_{o}\chi_{e}\vec{E}$$
where χ_e_ is the electric susceptibility, which is a dimensionless property that measures the polarizing ability of the material, and ε_o_ is the dielectric constant of the material. There are four types (Figure 1) of polarization mechanisms: (a) induced or electronic polarization, (b) atomic polarization, (c) orientation polarization, and (d) interfacial or space-charge polarization [41]. Here, electronic and atomic polarizations are resonance effects that are independent of temperature, whereas orientation polarization and space-charge polarization are relaxation effects that mainly depend on temperature [40].

Frequency is an important external factor that influences the dielectric constant. All four polarization modes present in a material operate in different frequency ranges [42]; although all of them respond and interact at low frequencies, individual polarization modes cannot interact at high frequencies. Consequently, the dielectric constant of a material gradually decreases with increasing frequency. In contrast, temperature positively influences the dielectric constant of the material, with the dielectric constant increasing with increasing temperature, which is ascribable to an increase in energy and material polarization. Humidity also influences the dielectric properties of a polymeric material, with the dielectric constant increasing with increasing humidity, which is ascribable to the formation of a water film on the material surface. 

#### 3.2.2. Resonating Polarizations

Resonating polarizations include electronic and atomic polarizations and are the main polarization modes that affect dielectric properties. Electronic polarization occurs when the electron cloud surrounding an atom is displaced with respect to the nucleus in an external electric field. A significant amount of energy is required to excite and establish resonance at electric field frequencies between 10^8^ and 10^10^ MHz (UV–visible range) because nuclei and electrons interact strongly [40]. In contrast, atomic polarization results from molecular skeletal deformations in an external electric field, which is the dominant polarization mode that affects the dielectric constant, despite its smallness (only one-tenth of the magnitude of electronic polarization); it appears at infrared frequencies between 10^4^ and 10^6^ MHz [42].

#### 3.2.3. Relaxing Polarizations

Orientational and space-charge polarizations have smaller effects and are considered to be relaxing polarizations. Orientational polarization is observed in covalent molecules with permanent dipole moments, where dipoles are aligned with the electric field [40]. This type of polarizability typically depends on the temperature and decreases with increasing temperature; it is excitable at radio frequencies in the 10^−9^ to 10^−6^ MHz range. However, more time is required to excite orientational polarization due to self-rotational resistance. Space-charge polarization or interfacial polarization is generally observed in multiphasic materials and is excitable at electric-field frequencies in the 10^−9^ to 10^−4^ MHz range [42], where the uneven distribution of two charges in the interior of the material results in a macroscopic dipole moment owing to the constrained free charge at the phase interface.

Accordingly, combining the abovementioned four effects leads to the following relationship between the dielectric constant ε and molecular polarizability ε: (5)$$\varepsilon =1+\frac{3N\alpha }{{3\varepsilon }_{o}-N\alpha }$$
where ε_o_ is the absolute dielectric constant (dielectric constant under vacuum; 8.85 × 10^−12^ F/m). 

Equation (5) reveals that molecular polarizability mainly influences the dielectric constant of a polymeric material. Consequently, a low-D_k_ PI material can be achieved using two strategies: (1) by reducing PI-chain polarizability using fluorine-containing monomers and (2) by reducing the packing density of the PI molecular chains by increasing the gaps (free volume) between the chains through the introduction of bulky groups or by engineering a porous PI structure. 

#### 3.2.4. Low D_k_ and low D_f_ Properties

With the rapid development of communication technology, the wavelength of communication waves has decreased [43]. However, high-frequency waves suffer from high-attenuation problems. Therefore, it is very essential to develop low-loss materials for high-speed signal transmission. To meet this requirement for high-performance low-dielectric materials, suitable mechanical and thermal properties are also very essential [44,45]. 

#### 3.2.5. Low Water Absorption

The absorption of water and moisture evidently affects the dielectric properties of PI materials because water has a large dielectric constant of approximately 80 [46,47]. Consequently, water absorption by the bulk material, or the formation of a water layer on the surface of a PI film, decreases its efficiency for low-k applications. Consequently, a low-D_k_ PI material is required to absorb water poorly. 

#### 3.2.6. Optical Transparency

The degree of film coloration and optical transparency are measures of the colorlessness and transparency of materials, usually evaluated in terms of the yellowness index (YI), total light transmittance (T_tot_), and haze index (turbidity) [36,48]. Although there are several criteria for completely colorless films, such as T_400_ (% transmittance at 400 nm), T_450_ (% transmittance at 450 nm), and T_500_ (% transmittance at 500 nm), T_400_ is usually the most reliable criterion. Hence, the YI is determined by measuring light transmittance at 400 nm using a UV–visible spectrophotometer in accordance with the ASTM E 313 protocol, and YI and haze index values for ~20–30 μm thick films should be below 3.0 and 1.0, respectively [48]. For simplicity, films with high T_400_ values (>80%) typically have YI values of less than 3.0. 

The extent of coloration and turbidity of a PI film usually depend on chemical factors, such as charge-transfer (CT) interactions and electronic conjugation in the PI chain, which are indicative of the nature of the PI backbone [49]. However, the YI is adversely affected by the thermally or light-induced partial decomposition of terminal amino groups, the thermally induced degradation of aliphatic units in the main chain or side groups associated with high-temperature processing, and occasionally, the type of solvent used during processing. 

A PI with a sufficiently high molecular weight has a low number of amino end groups, whereas a low-molecular-weight PI has a sufficient number of unstable amino end groups that colorize the PI film via thermally/photo-induced decomposition [49], which can be prevented by end-capping the free amino end groups using monofunctional reagents, such as phthalic or acetic anhydride. PIs containing aliphatic units (in the main chain or side groups) need to avoid heat and aerobic conditions owing to the abrupt coloration observed at higher temperatures and the presence of air during the film-making process. However, monomer purity mainly affects the color of the PI film; consequently, the purification of the amine and anhydride using suitable methods, such as recrystallization with an appropriate solvent or sublimation under reduced pressure, is quite important, with the latter being an effective method for decolorizing monomers. 

Finally, the imidization method also affects the color of the film. Three main methods are used to prepare PI films (as shown in Figure 1). Although highly soluble PIs can be synthesized by any of the three methods, chemical imidization (at room temperature) using monofunctional dehydrating agents, such as acetic anhydride (Ac_2_O), is the best option because it avoids high-temperature processes in which degradation can colorize the product [50]. However, the preparation of an insoluble PI film is confined to the conventional two-step imidization process, in which films are prepared by thermal imidization after casting the PAA solution into glass molds or onto slides. 

Although the degree of coloration of a PI mainly depends on the chemical structure of the PI and the imidization method, technical factors, such as the type of solvent used during film preparation and the processing conditions, are also important for obtaining a colorless PI (CPI) film. Finally, the choice of solvent is also an important factor for attaining optically transparent PI films, as discussed in detail in our previous review [35]. Even though NMP is said to be a preferred solvent for preparing PAA by the two-step method, it can colorize the film during the film-making process at elevated temperatures (second step) because unknown colored products are formed via the oxidation of NMP, which is a disadvantage. In contrast, GBL is the best solvent for use in the film-preparation step (either thermal or chemical imidization) despite its high boiling point, owing to the combined effects of moderate dissolvability, a lower degree of coloration due to oxidative stability, and the ease of evaporation during film formation. 

CPIs are very important in various applications, such as optoelectronic devices, active-matrix organic light-emitting display devices (AMOLEDs), touch sensors, printed circuit boards (PCBs), thin-film transistors (TFTs), and solar cells [51,52]. Furthermore, the AMOLED displays of mobile electronic devices, such as smartphones and tablet PCs, are expected to be continuously improved in terms of visibility, durability, weight, and flexibility; consequently, CPIs have attracted significant amounts of academic and industrial research attention [53,54]. Recent research has demonstrated that adding specific structures to the main chain of PIs, such as fluorinated groups, alicyclic-group-containing monomers, noncoplanar structures, meta-substituted structures, or sulfones, can increase the transparency of PI films while lowering CTC formation [55]. 

Based on the aromatic nature of the main chain, the CPIs can be classified as aromatic and nonaromatic CPIs, while Wenlin et al. classified colorless PIs into fluorinated CPIs, alicyclic CPIs, noncoplanar CPIs, and other CPIs based on the main-chain composition (Figure 2) [35,55]. 

Fluorinated CPIs are an important class of CPIs, and most of the CPIs found in the literature are based on fluorinated PIs. The electronegative fluorine atoms added to PI chains suppress the CTC, which enhances both dielectric and optical properties [55]. Conversely, the fluorine atom, as in the -CF_3_ group inclusion, may result in a decrease in mechanical strength and glass transition temperature (T_g_), despite improvements in optical and dielectric characteristics. 

Alicyclic CPIs are a class of CPIs in which nonaromatic monomers, like diamines, dianhydrides, or both, are used. Alicyclic CPIs also reduce the CTC effect by preventing extended conjugation in the main chain. Additionally, this has a considerable effect on the mechanical and thermal properties. As a result, it is crucial to consider the extent of alicyclic structure without significantly compromising the mechanical and thermal characteristics. 

As plastic film substrates, CPIs are broadly applicable to future microelectronic devices, such as flexible display devices, touch panels, solar cells for photovoltaic applications, and antennas for communication devices. Furthermore, a CPI provides signal transmission pathways and a medium for communication applications while also providing structural support [10]. However, the practical applications of a PI are mainly dependent on its heat resistance and thermal and dimensional stability. Here, the thermal stability of a PI is mainly determined by the processing temperature, whereas its optical and mechanical properties are adversely affected by high processing temperatures. Therefore, the optimal processing temperature that maintains both thermal and optical properties needs to be determined. Another important aspect of a PI, dimensional stability, depends on the CTE because expansion causes serious problems, including the misalignment and adhesion failure of various microcomponents, laminate warpage, and transparent electrode breakdown during device fabrication and/or during operation [56,57,58].

#### 3.2.7. Packing Factor (d-Space) and Dimensional Stability

Polymer-chain packing in a film generally depends on several factors that enforce competition, including the rigidity of the main chain, rotational barriers that obstruct main-chain rotations, and intermolecular forces [59]. Close molecular packing is related to D_k_ stability and low D_f_ values over a range of frequencies, where the rigidity of the PI backbone has a positive effect by reducing dipole deflections in the electric field [28]. PI-chain stacking reduces D_k_ by decreasing the polarizability of the PI film. 

The aggregate structure of a PI film, including the molecular packing and orientation of its polymer chains, plays an important role in determining the dielectric properties of the material. Molecular disorder induced by the tetrahedral carbons in the PI polymer chains and the pendant side groups is important for molecular packing, which can be determined for an amorphous glassy polymeric material using wide-angle X-ray diffractometry (WAXD), in which prominent peaks are used to calculate the d-spacing according to the Bragg equation [59]. 

Dimensional stability is an important consideration in microelectronic devices. Due to the shrinking of electronic devices and the forced integration of many components in PCBs brought on by technical advancements, these devices may heat up and experience thermal expansion [19,60]. Additionally, the board material may be impaired during fabrication processes, like the soldering and drilling of PCBs. To withstand this, the interlayer material or board material should have great mechanical stability and low thermal expansion [60]. 

## 4. Structural Designs of Transparent PI Films with Low Dielectric and Low Water-Absorbing Properties

### 4.1. Polyimide Synthesis, Characterization, and Properties

Polyimides form a family of polymers that are generally synthesized via polycondensation reactions involving diamines and tetracarboxylic acid dianhydrides in extremely anhydrous high-boiling-point polar aprotic solvents, such as dimethyl sulfoxide (DMSO), dimethylformamide (DMF), dimethylacetamide (DMAc), *N*-methyl-2-pyrrolidone (NMP), γ-butyrolactone (GBL), and m-cresol [35,61]. The solvent is chosen based on requirements that include dissolvability, toxicity, ease of evaporation, oxidative stability, and hygroscopicity. 

A variety of methods for synthesizing different polyimides have been developed over the past few decades [62], with the general synthetic route shown in Figure 2. Two main routes are used to synthesize polyimides, namely, a one-step method and the classical two-step polymerization method. The latter first produces polyamic acid (PAA) by reacting an equimolar mixture of a diamine and a tetracarboxylic acid dianhydride (Figure 3), which is a rapid exothermic process involving the nucleophilic addition of the amine at the electrophilic carbonyl carbon of the anhydride, which then cyclodehydrates to form the strong imide backbone of the PI via the removal of water either through the use of a chemical dehydrating agent (chemical imidization, Figure 1a) or at elevated temperatures (thermal imidization, Figure 1b). In contrast, complete PI cyclization is achieved directly in a high-boiling tertiary amine (e.g., isoquinoline) at elevated temperatures using an equimolar mixture of a diamine and a tetracarboxylic acid dianhydride in a one-step method (Figure 1c), in which water is continuously removed from the reaction mixture by azeotropic distillation.

Molecular weight is a crucial factor that affects the processibility of a PI film [63]. Consequently, either the stoichiometric imbalance method (excess of one reactant over the other) or the monofunctional reagent method (addition of a monofunctional reagent to the reaction mixture) is required to control the molecular weight [35]. 

### 4.2. Examples of Structural designs of Transparent PI Films with Low Dielectric and Low Water-Absorbing Properties

Although PIs are attractive for use in many areas owing to their unique properties, their structure–property relationships need to be better understood in order to improve their characteristics and advance their use in new commercial applications. A PI can potentially be used as a dielectric-layer material as well as a flexible transparent substrate in an integrated device. Meta-substituted and trifluoromethyl structures are used to improve transparency, decrease yellowness, lower the dielectric constant, and maintain the comprehensive properties of a polyimide film. The two-step method has been used to synthesize highly transparent PI films with low k values. 

Highly transparent, colorless, and low-D_k_ PIs were developed by introducing meta- trifluoromethyl substituents into phenylenediamine-based (PDA-based) PIs [64]. A highly transparent, low-k CF_3_-m-PDA-FDA polyimide film was synthesized from m-CF_3_-m-PDA and 4,4′-(hexafluoroisopropylidene)diphthalic anhydride (FDA). The CF_3_-m-PDA-FDA film met the requirements of high heat resistance (T_g_ = 296 °C) and good thermal stability, along with suitable mechanical properties, for use in high-temperature microelectronic applications.

Balasubramanian et al. developed a series of poly(ether imides) (PEtIs) based on 3,3’,4,4’-benzophenonetetracarboxylic dianhydride (BTDA) or pyromellitic dianhydride (PMDA) and four diamines with different aliphatic carbon-chain sequences ((CH_2_)_n_, n = 3–6) [65]. The dielectric constant is well known to be closely related to repeating-unit polarizability, which is determined by the chemical structure of the PEtI. Consequently, increasing the chain length led to a decrease in both the dielectric constant and dielectric loss in both series owing to a decrease in polarizability and polarity dilution (of the imide ring) by the phenylene ether units of the diamine; however, it negatively affected thermal and mechanical properties. 

Alicyclic fluorinated PI (FPI series) films were prepared using 1,3-bis((4-amino-2-(trifluoromethyl)phenoxylmethylene)-1,2,2-trimethylcyclopentane (BAFMT), a new camphor-derived diamine, and systematically compared to the API series [66]. The FPI series of PIs exhibited superior properties in terms of optical transparency, water absorption, surface energy, and solubility in organic solvents owing to the bulky fluorinated trimethyl-substituted cyclopentyl structure in the main chain. However, a small decrease in thermal stability (compared to the API series) was observed, which is ascribable to the alicyclic groups, although reasonable values of T_g_ and 5% weight loss temperature (T_d_^5%^) were reported. 

Alicyclic diamines and dianhydrides derived from 4,4’-isopropylidenedicyclohexanol (IDH; cis-hydrogenated bisphenol-A) were used to develop new colorless organic-soluble PI films using the thermal imidization method (Figure 3) [67]. These PIs were exceptionally highly soluble, with PI-2, PI-5, and PI-6 even soluble in low-boiling solvents, such as chloroform and dichloromethane, which is attributable to fewer intermolecular interactions and looser molecular chain packing due to the IDH moieties, as confirmed by the d-space values obtained by X-ray diffractometry. The thermal and mechanical properties of these materials are also affected by the percentage of aliphatic moieties in the PI films. T_g_ was observed to increase with decreasing aliphatic percentage in the PI, with values of 221, 233, and 241 °C determined by dynamic mechanical analysis (DMA) for PI-6 (52%), PI-5 (29%), and PI-7 (0%), respectively. However, different T_g_ values were determined by differential scanning calorimetry (DSC) for the same PIs, namely, 219, 248, and 239 °C, respectively. In both cases, the semi-aromatic polyimides (semi-APIs) exhibited comparable T_g_ values to those of aromatic PI-7, with the real value mainly dependent on the conformational rigidity of the polymer, in which the rigidity of a single-molecule chain is dictated by the rigidity of its repeating units. Consequently, altering the molecular packing by introducing IDH moieties increases processibility due to improved solubility in low-boiling solvents. Furthermore, these PIs exhibit low dielectric constants, low water absorbability, and good transmittance and are, therefore, suitable for use in optoelectronic applications.

Recently, two new pyridine-containing diamines were synthesized using a facile method, with organic-soluble fluorinated polyimides synthesized using various dianhydrides. The prepared films were flexible and tough and exhibited UV–visible cut-off wavelengths in the 342–393 nm range (Figure 4) [68]. However, DSC studies revealed that these PIs have moderate T_g_ values (239–306 °C) and 5% weight loss temperatures of 498–523 °C and 449–511 °C under N_2_ and in air, respectively. Thus, these PIs are more thermally stable and exhibit adequate transparency along with good mechanical and low-k properties (D_k_ values of between 2.71 and 2.92 at 1 MHz). Although PPI-4–PPI-6 (Figure 3) are structurally similar (they all have FDA in common), their dielectric constants differ owing to differences in fluorine content, the presence of pyridine, or both. PPI-4 has the lowest dielectric constant (D_k_ = 2.71) among the less-fluorine-containing PPI-5 and non-pyridine-containing PPI-6. Here, the higher D_k_ values of the non-FDA films are attributable to the strong polarizability of the PI backbone but are nevertheless superior to that of commercial PI Kapton^®^ (D_k_= 3.67 at 1 MHz). However, the introduction of bulky trifluoromethyl moieties and strongly electronegative nitrogen atoms into the amine monomers led to less-polarizable repeating units with higher fractional free volumes (FFVs) in their PI chains and low dielectric constants as a consequence. 

In a similar manner, Dong et al. developed soluble CPI films based on 4- and 6-methylated pyridine-containing diamines (Figure 3) [69]. These PIs have thermal properties that are comparable to those of PIs previously reported in [68] and exhibited moderate T_g_ values (262–275 °C) with DSC and 5% weight loss temperatures of 468–499 °C and 500–524 °C under N_2_ and in air, respectively. However, their optical transmittance and dielectric properties were further improved (optical transmittance ≥90% at 450 nm and D_k_ > 2.52 at 1 MHz) for use in optoelectrical applications.

Semi-alicyclic optically transparent PIs based on bicyclo [2.2.2]oct-7-ene-2,3,5,6-tetracarboxylic dianhydride (BODA) and various diamines containing ortho-substituted methyl and substituted phenyl pendant groups were developed for low-k applications, and their properties were compared with those 1,4-cyclohexanedicarboxylic acid (CHDA) (Figure 5 and Figure 6) [70]. The PIs were prepared via one-step imidization in NMP at 180 °C, followed by solution casting; their T_g_ values were determined by DMA to lie between 336 °C and >400 °C, with 5% weight loss temperatures of 421–501 °C determined by thermogravimetric analysis (TGA) under N_2_. Although these PI films show superior thermal and mechanical properties and are highly processable and very soluble, they exhibit inferior dielectric properties, optimal optical transparency, and water-absorbing capabilities to those of commercial PIs. The same authors also reported the development of highly transparent semi-aromatic PI films based on CHDA and various fluorinated aromatic diamines, along with 4,4’-oxydianiline (ODA) (Figure 5) [71]; these films exhibited T_g_ values (DSC) in the 275–370 °C range and 5% weight loss temperatures of 436–541 °C under N_2_ (TGA). Subsequently, Zhang et al. developed BODA-based semi-alicyclic PIs with various diamines bearing trifluoromethyl groups for low-k applications and compared their properties to those prepared using 1,2,3,4-cyclobutanetetracarboxylic dianhydride (CBDA) (Figure 6) [63]. The T_g_ values (DSC) of these PIs ranged between 285 and 390 °C, with 5% weight loss temperatures of 405–440 °C under N_2_. These PI films exhibited moderate thermal and mechanical properties but were highly processable owing to their excellent solubility; however, they exhibited inferior dielectric properties and water-absorbing capacity compared to those of commercial PIs. 

1,4:3,6-Dianhydro-d-glucidol-based fluorinated organic-soluble PIs were developed using the two-step thermal imidization method from semi-aromatic 1,4:3,6-dihydro-d-glucidol diamines (DGDAms) (1a, 1b, and 1c in Figure 7, top), various dianhydrides, and FDA (Figure 7) [72]. The prepared PI films were flexible, tough, and soluble in common polar solvents. Furthermore, the films exhibited good transparency (72–87%) at 450 nm, low water absorbability, and exceptionally low dielectric constants (2.02–2.40 at 1 MHz). All PIs exhibited moderate T_g_ values (207–280 °C) and 5% weight loss temperatures of 427–536 °C and 404–522 °C under N_2_ and in air, respectively. Thus, these PIs have good thermal properties in terms of heat and flame resistance. Structurally similar PIs (PI-7, PI-6, and PI-5) exhibited similar transmittance and hydrophobicity (low water absorbability), with discrepancies in the dielectric constant (PI-6 exhibited a slightly higher D_k_) attributable to the electron-donating nature of the methyl groups in amine 1b. 

The same research group subsequently developed transparent, soluble PIs from dianhydrides based on 1,4:3,6-dianhydro-d-mannitol and various aromatic diamines using the two-step thermal imidization method (Figure 7) [73]. The first four PIs (DMDAn-1 to DMDAn-FDA) were prepared from aromatic diamines and 1,4:3,6-dihydro-d-mannitol dianhydride (DMDAn), whereas the others were prepared from diamines (DGDAm-1, 2, and 3) published in [72]. All PIs were optically transparent, soluble in polar organic solvents, and sufficiently tough and flexible. PIs 4–7 exhibited enhanced solubility and transmittance compared to the others, which is attributable to their aromatic trifluoromethyl units, 1,4:3,6-dihydrohexitol units, or both. PI-4 was less polarizable owing to the FDA units, whereas the formation of a CTC contributed to superior optoelectrical properties. In contrast, both the trifluoromethyl and alicyclic contents in PI-7 play key roles. However, the improved solubility and transmittance of PI-5 and PI-6 and their moderately lower D_k_ values are attributable to looser chain packing, which is ascribable to weaker interchain interactions associated with the alicyclic units.

Highly thermostable low-k poly(ether-imide) (PEtI) films and their silica nanocomposites were developed using 9,9-bis [4-(3,4-dicarboxyphenoxy)phenyl]fluorene dianhydride (FDAn) (as a fluorene-containing dianhydride) and various diamines (Figure 8) [74]. All prepared films exhibited moderate DSC-determined T_g_ values (205–225 °C). Moreover, the PEtIs films were highly thermally stable and exhibited T_d_^5%^ values in the 500–535 °C range in air. The films had exceptionally low dielectric constants (1.96–2.83) at 10 kHz, where structural changes in the PI backbone that increased the flexibility, packing, and polarization of repeating units affected the *D*_k_ values of FDAn-1–3. To further reduce the D_k_ (up to 2.44) value, Wu et al. modified the fluorene moieties of ODPA- and FDA-based polyimides by directly silylating the pendant hydroxy groups of 9,9-bis(4-aminophenyl)-2,7-dihydroxy-fluorene (Fn1) and 9,9-bis(3-methyl-4-aminophenyl)-2,7-dihydroxy-fluorene (Fn2) with tert-butyldimethylsilylchloride (Figure 8) [75], which also led to enhanced thermal properties (T_g_ > 311 °C; T_d_^5%^ > 469 °C). The same group also reported similar results using tert-butyldiphenylsilylchloride (Figure 8) [76]. 

Nakagawa and Morikawa developed aromatic PIs (PI-1x and PI-2x) from Z-shaped linear sexiphenyltetracarboxylic dianhydrides and various aromatic diamines [77,78]. PI-1x-type polyimides exhibited high D_k_ and low T_g_ values (232–280 °C) and were insoluble in organic solvents even when heated. In contrast, the PI-2x PIs showed high T_g_ values (270–345 °C) and low D_k_ values and were highly soluble. The same group subsequently developed PIs based on tetraphenylnaphthalene units [79], which showed moderate T_g_ values (270–315 °C) and low dielectric constants (2.72–2.9). Another study reported PIs based on 1,4-bis [4-(4-aminophenoxy)phenyl]-2,3,5,6-tetraphenylbenzene, a diamine-bearing hexaphenylbenzene, and various aromatic dianhydrides [80]. The water absorbability and *D*_k_ values of the members of the PI-2x series decreased as the number of phenyl rings increased, with the lowest values observed for FDA-containing PI-2g. 

With the aim of developing 5G/6G communication materials, Zhang et al. synthesized liquid-crystal poly(ester-imide) (LCPEI) PEI films based on various fluorinated imide dicarboxylic acids by copolymerizing different moieties containing carboxylic and phenolic functionalities using the direct-esterification method reported by Higashi (Figure 9) [81,82]. The prepared films exhibited moderate T_g_ values (196–220 °C), good thermal stability (T_d_^5%^ > 450 °C), and suitable mechanical strength, which are ascribable to the nature of the PI backbone structure; these properties (T_g_: 164–238 °C; T_d_^5%^ > 450 °C [82]) were unaffected by the introduction of various amounts of fluorine into the bulky pendant groups. However, the solubility of the PEIs in general organic solvents, such as chloroform and DMF, facilitated easier processing. 

Furthermore, LCPEIs produced via solution polymerization exhibited advantageous properties compared to those produced by other methods, including tunable imide contents in the polymer backbone and facile processing, compared to Kuraray’s Vecstar^®^ film-preparation method; they also have better (i.e., lower) k values (D_k_ = 3.07 and 2.91) than Kapton (D_k_ = 3.45) and Vecstar^®^ (D_k_ = 3.30) marketed by DuPont due to the higher free volume ascribable to the bulky phenyl or four methyl groups in the imide repeating unit [81]. In addition, PEI-6F25AF (D_k_ = 2.60) and PEI-6FD25AF (D_k_ = 2.79) exhibited even lower D_k_ values when various numbers of electron-withdrawing fluorine atoms were introduced onto the bulky pendant groups in the LCPEI chains [82]. 

A PI containing 1,2-diphenyl pendant groups was developed by Lee et al. using novel ester-containing diamines and FDA [1]. The developed PI was highly soluble in polar organic solvents and exhibited excellent optical transmittance (T_450_: 85.57% at 450 nm). Additionally, it showed a very low dielectric constant (2.17 at 1 MHz). Qi et al. developed ester dianhydrides based on alicyclic cis-ethylenic double-bond-containing bicyclo [2.2.1]-heptyl-based dianhydride (BHEM) and its aliphatic cis-ethylenic double-bond-containing dianhydride (BHSM) counterpart. PI films were processed with three different diamines, and their properties were compared [83]. Despite the prepared PIs showing sufficient physicomechanical properties, some differences, such as aggregation, thermal, mechanical, optical, conductivity, and dynamic dielectric properties, were observed due to disparities in the polarity and polarizability of the PI-chain repeating units. WAXD revealed that the BHEM-based PIs aggregated more strongly through interchain π–π stacking via their sp^2^ configurations. Moreover, the higher polarity of these monomer types endows these PIs with strong CTC behavior and low dielectric constants, thereby imparting them with higher T_g_ values and stronger electrical conductivity. In contrast, the low-polarity, bulkier, and flexible sp^3^ configuration of BHSM led to low-dielectric films by inhibiting interchain CT interactions. 

Ester-linked fluorene-containing PEIs constructed using various diamines, including 4,4’-methylenebis(cyclohexylamine) (MBCHA), 9,9-bis(4-hydroxy-3-methylphenyl)fluorene dianhydride (BMPFLDAn), and 9,9-bis(4-hydroxyphenyl)fluorene dianhydride (BPFLDAn), were highly heat-resistant and exhibited good photosensitivity (Figure 10) [84]. The T_g_ values of the prepared PEIs sometimes exceeded 300 °C (mostly ranging between 259 and 356 °C), and CHDA-based PEIs were highly optically transparent and very soluble in common organic solvents. Furthermore, CPIs containing small amounts of fluorene-containing units were photosensitive and produced positive-tone patterns when performing photolithography. The refractive-index-based D_k_ values of the PIs were not significantly influenced by water absorption, with directly measured D_k_ values of 2.87–3.07 at 1 MHz and 2.75–3.02 at 10 MHz. 

Trifluoromethyl-containing transparent poly(amide–imide) (PAI) films were developed using two different (centrosymmetric and/or noncentrosymmetric) biphenyl-based amines and an alicyclic trimellitic anhydride (1,2,4-cyclohexanetricarboxylic anhydride) (Figure 11) [85]. All PAI films were highly transparent owing to the combined effect of the alicyclic moieties and pendant trifluoromethyl groups. Furthermore, these PAI films were thermally stable (T_d_^5%^ > 418 °C under N_2_; > 381 °C in air) and thermo-resistive (T_g_ > 304 °C) in nature due to their close-packed polymer chains, as evidenced by their grazing-incidence wide-angle X-ray scattering (GIWAXS) patterns.

Two series of PAIs bearing various pendant groups were recently developed based on two novel diamine diacid monomers (Figure 11) [86]; these PAIs were highly thermally resistant (T_g_ > 232 °C), thermally stable (T_d_^5%^ >493 °C), and soluble in a variety of organic solvents, which is ascribable to their bulky pendant groups. Furthermore, they exhibited good optical transmittance and D_k_ values (<2.98) suitable for communication applications. Lee et al. synthesized transparent nanoporous PAI films for low-dielectric applications [87] from isophthalic dihydrazide, FDA, and silica nanoparticles using the two-step thermal imidization method, followed by silica etching with a 10% HF solution. Films with silica nanoparticles and nanoporous structures following silica removal were studied using various characterization techniques; DSC and TGA revealed that the physically modified films showed improved heat resistance (T_g_ >250 °C) and were less thermally degradable (T_d_^5%^ > 364.97 °C). However, the YI value decreased when moving from PAI-0 to PAI-1 (2.31 vs. 1.09) owing to the porous nature of these materials, despite the transmittance gradually decreasing. Furthermore, D_k_ was also observed to decrease from 3.35 for the non-porous film to 2.51 for the nanoporous PAI-1 film (HF-etched film loaded with 1% silica nanoparticles). 

Phosphine-oxide-based diamines and their PIs (DP-PIs) were developed for low-dielectric applications, and their structure–property relationships were studied by varying their flexible main-chain moieties, which include ether, carbonyl, isopropylidene, and hexafluoroisopropylidene linkages (Figure 12) [88]. The thermal properties of the prepared DP-PIs were studied by DSC and TGA, which revealed moderate T_g_ values (216–271 °C) and 5% weight loss temperatures of 473–487 °C and 467–486 °C under N_2_ and in air, respectively. However, the keto-linked DP-PI exhibited the highest D_k_, followed by the ether-linked, isopropylidene-linked, and hexafluoroisopropylidene-linked DP-PIs. The D_k_ values of the prepared DP-PIs are comparable to that of Kapton^®^ and varied according to the flexibility, bulkiness, and polarity of the repeating unit. Polyimides based on diamines bearing phosphene-oxide and fluorine groups were developed in another report for adhesive and low-dielectric applications (Figure 12) [89]. In addition to the excellent thermal properties in terms of heat resistance (T_g_ > 276 °C) and thermal stability (T_d_^5%^ > 500 °C in air), these TP-PIs also exhibited low *D*_k_ values (2.34–2.89) suitable for use in communication applications. 

Kuo et al. studied the structure–dielectric-property relationships of PIs with various functional groups by preparing a series of 36 PIs and further correlated their structural parameters, such as fluorine content (F%) and volume polarizability (P/V), with *D*_k_ and their imide-group content (imide%) with D_f_ (Figure 13) [90]. D_k_ correlated with F% and P/V, with correlation coefficients of 0.98 and 0.90, respectively, and highly correlated (0.95) with imide%. The good correlation between D_k_ and P/V is possibly ascribable to the contributions of electronic and atomic polarizations to D_k_ at high frequencies. Moreover, the Clausius–Mossotti equation was found to largely govern D_k_. Finally, these researchers also showed that fluorine-based PIs could deliver both low D_k_ and D_f_ values, which are essential for 5G/6G communication applications.

An organic-soluble, colorless, partially alicyclic polyimide based on Troger’s base (TB) was developed for multiple applications, including low-dielectric and solar-energy applications, as it fluoresces/phosphoresces at long wavelengths [91]. The prepared material exhibited a D_k_ value of approximately 2.66, which is superior to those of commercial PIs, although this value is slightly higher than that of the corresponding aromatic TB-PI based on FDA (D_k_ = 2.57) [92]. Moreover, transmittance at 400 nm was greatly enhanced owing to its less polar alicyclic moieties; this PI was also highly mechanically and thermally stable. Moreover, the PI exhibited dual fluorescence and phosphorescence emissions at low temperatures, which is useful for spectrally converting solar UV radiation into visible light. 

Novel organic-soluble polyimides and copolyimides (co-PIs) were synthesized based on 4,4′-(4,4′-isopropylidenediphenoxy)bis(phthalic anhydride) and triphenylmethane-based diamine/4,4’-oxydianiline (Figure 14) [59]. The co-PI films exhibited lower D_k_ and *D*_f_ values compared to the PI films, along with adequate toughness and good mechanical and thermal properties. The d-spacing decreased from 5.33 to 4.95 in moving from PI-1 to PI-3 due to more densely close-packed polymer chains resulting from various factors, such as strong electronegativity (polar interchain interactions) in the case of PI-2 and restricted molecular polymer-chain rotations due to the larger Br atoms on the phenyl ring in the case of PI-3. In contrast, the packing ability of co-PI-1 and co-PI-2 was slightly better than that of co-PI-3. 

A series of low-dielectric 3,3’,4,4’-biphenyltetracarboxylic dianhydride (BPDA)/p-PDA/2,2’-bis(trifluoromethyl)benzidine (TFMB) co-PI fibers were prepared using a two-step wet-spinning method, which resulted in enhanced properties, including mechanical strength and thermal stability, along with a low dielectric constant (2.48 at 10G Hz), through enhanced micro-structural interactions facilitated by the introduction of trifluoromethyl-containing TFMB moieties into the BPDA/p-PDA backbone [93]. Experimental and molecular dynamics simulation data suggest that the TFMB moieties on the BPDA/p-PDA PI chains greatly influence the charge distribution, chain flexibility, and structure of the aggregated PI fibers. Furthermore, the higher FFV, which is ascribable to a higher proportion of TFMB in the PI fibers and is associated with looser polymer-chain packing and the low electronic polarizability of the CF_3_ group, led to a lower dielectric constant (2.48 vs. 3.0) at 10 GHz. 

Partially aliphatic copolyimides have been developed using structurally rigid cycloaliphatic and aromatic dianhydrides and ODA [94]. With the exception of BPDA-10, the prepared films are flexible, with WAXD revealing that most films are amorphous with ill-defined patterns, whereas some are crystalline owing to the highly aromatic nature of the PI films. In contrast, all 90% cycloaliphatic dianhydride or rel-(1′R,3S,5′S)-spiro[furan-3(2H),6′-[3]oxabicyclo [3.2.1]octane]-2,2′,4′,5(4H)-tetrone (DAn) films are highly transparent and show transmittance that exceed 90% at 400 nm. Furthermore, these films exhibit refractive-index-based dielectric constants in the 2.5–2.65 range and are, therefore, well suited for use in low-dielectric applications. 

A series of co-PIs based on a new diamine and 1,4-bis(4-amino-2-trifluoromethylphenoxy)benzene (BATB) (bearing a pendant 2,6-biphenylpyridyl moiety) and FDA (bearing (4-trifluoromethyl)phenyl pendant groups) were prepared using the two-step thermal imidization method [95]. The T_g_ value increased, while the D_k_ value remained at > 3 as the amount of the 2,6-biphenylpyridyldiamine was increased, which is essential for better communication speed. The prepared co-PIs were amorphous and exhibited superior thermal resistivity (T_g_ > 272 °C) and stability (T_d_^5%^ > 520 °C) and are highly processable owing to their superior solubility. 

Li et al. fabricated co-PIs based on BPDA/PDA/2,20-bis-(trifluoromethyl)-4,40-diaminobiphenyl (TFDB) and studied their structure–property relationships with respect to the number of trifluoromethyl groups [96]. Experimental and computational studies revealed that D_k_ decreases with increasing TFDB content in the PI owing to the incorporation of more -CF_3_ groups in the PI, which enhances both the FFV (17.94–20.52%) and polarizability. Consequently, 70% TFDB content in the co-PI film was sufficient to ensure adequate mechanical strength and a *D*_k_ value of 3.03; this value is comparable to that of an FDA-ODA PI film (D_k_ = 3.0). 

Fluorinated co-PI films based on BPDA and FDA and TFDB diamines were also synthesized for dielectric applications [28]; these films showed stable dielectric constants and low *D*_f_ values over a range of frequencies (10–60 GHz) that are attributable to limited dipole deflection in the electric field associated with the close-packed rigid polymer backbones. Furthermore, the relationship between the molecular structure and dielectric properties was described in terms of the equilibrium structures of the repeating units in the films based on FFV values calculated from molecular dynamics simulations. 

A series of porous co-PI films with ultralow dielectric constants and low water absorptivity were developed by Lv et al. [97]; these PI films exhibited good thermal (T_g_ = 300 °C; T_5%_ = 411 °C) and mechanical properties. Bulky adamantane groups, polydimethylsiloxane (PDMS) segments, and nanopores obtained via the thermal dissociation of poly(ethylene glycol) (PEG) led to ultralow dielectric properties (D_k_ = 1.85; D_f_ = 0.012 at 1 MHz), which are important for ensuring superior performance in interlevel insulation and 5G/6G communication applications. Recently, Miyane et al. developed thermally stable and colorless co-PIs for potential use in transparent organic electronic device applications, such as organic field-effect transistors. These PIs exhibit low D_k_ and D_f_ values, ultralow CTEs, and sufficient mechanical durability [5]. Moreover, the methyl groups of the PDMS segments endow these PIs with hydrophobicity; hence, they absorb far less water than Kapton^®^ (0.93–1.21% vs. 3.54%). Finally, these PIs exhibited excellent dielectric constants (2.84–1.85) at 1 MHz and showed almost constant D_f_ values in the broad 40 Hz to 1 MHz frequency range, which is beneficial for microelectronic applications. 

Table 1 summarizes the dielectric properties, water absorbability, and optical transparency of the PI films discussed in this review.

## 5. Conclusions and Future Perspectives

APIs are endowed with exceptional mechanical and thermal properties required for 5G/6G communication systems. However, they suffer from unwanted CTCs that result in optical inefficiencies. Incorporating side groups into rigid PI chains reduces unwanted inter- and intramolecular CTCs in the PI material but leads to a loss in modulus. Consequently, we suggest using small-to-medium side chains that affect API moduli to a lesser extent. 

The inclusion of side groups (aromatic side chains like phenyl, naphthyl, fluorene, or their substituted ones, silylated side chains, non-polar alicyclic moieties, aliphatic groups like -CH_3_ and -CF_3_) can affect the properties of a PI both positively and negatively; they can enhance the dielectric properties of the PI while altering its refractive index, anisotropy, and thermal and mechanical properties. For instance, the addition of side groups alters the dielectric and thermal characteristics of the DGDAm1a-FDA structure (D_k_ = 2.15; T_g_ = 280; T_d_^5%^ = 461). When methyl groups are introduced, the properties change to D_k_ = 2.17, T_g_ = 260, and T_d_^5%^ = 446 (DGDAm1b-FDA), whereas -CF_3_ groups are introduced at the same position, and the properties are D_k_ = 2.02, T_g_ = 260, and T_d_^5%^ = 445 (DGDAm1c-FDA). Hence, special care is required when selecting side groups to ensure a positive outcome. The incorporation of kinkable or flexible linkages also enhances the mechanical properties of the PI material, including its flexibility; however, its modulus mainly depends on the linkage type and length.

Semi-API films are generally highly transparent and appropriately thermally stable. The twisted structure (due to sp^3^-hybridized cyclic or acyclic moieties) and low polarizability of a semi-API lead to weak intermolecular interactions, with loose chain packing leading to low-k films. Furthermore, fluorine substitution usefully modulates the FFV and polarizability of the PI repeating unit; furthermore, it also increases PI processibility by increasing solubility and reduces *D*_k_ by decreasing water uptake. Fluorinated PIs benefit more from trifluoromethyl groups than from individual F atoms, which is ascribable to their greater hydrophobicity and lower polarizability, which is evident from the LCPEI-3F (3 F:D_k_ = 3.0), LCPEI-4F (4 F:D_k_ = 3.01), and LCPEI-6F (2 CF_3_:D_k_ = 2.93).

This review discusses the structural designs of PIs in terms of optical transmittance, water absorption, and thermal and dielectric properties. Furthermore, it covers different types of PIs with ether, ester, and amide linkages, along with acyclic or semi-acyclic moieties, with the aim of evaluating structure–property relationships involving optical and dielectric properties for communication applications. This information is intended to assist investigators when extending research to high-performance PIs for communication applications. 

The demand for low-D_k_ PI-based materials has gradually increased as communication technologies have advanced. PI-based materials are endowed with excellent properties for use in microelectronic devices in the forms of interlayer insulation and the optimal internal transmission time. Hence, enhancing the dielectric, optical, and other properties of PIs is essential for the further development of high-performance communication devices. High-performance PIs for 5G and 6G communication devices still face challenges, despite active research into low-dielectric PIs. The following key challenges need to be addressed to improve PI performance: 

The current pursuit of low-D_k_ and low-D_f_ materials leads to the loss of thermal (low T_g_) or optical properties; therefore, developing PIs endowed with all of the required properties represents new challenges in this field, such as solving low adhesiveness, cracks in the passivation layer, etc., during soldering, cutting, and punching.

Although the use of low-D_k_ PIs strengthens prospects for microelectronic communication devices, research in this area needs to progress further. Many obstacles need to be overcome in relation to obtaining PIs with appropriate molecular weights and synthetic yields. Hence, industrial-scale production should be an important focus.

Synthesizing novel monomers requires skilled researchers, with molecular yield often suffering due to a lack of research. Therefore, in addition to focusing on the synthesis of novel monomers, attention should also be paid to suitable copolyimides and organic–inorganic hybrid materials as a better way to address issues affecting thermal stability (low T_g_) and optical properties. 

## Data Availability

The data presented in this study are available on request from the corresponding author.

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
