# Peer review of "Structural Designs of Transparent Polyimide Films with Low Dielectric Properties and Low Water Absorption: A Review"

_nanomaterials, 2023, doi:10.3390/nano13142090_

Round 1
Reviewer 1 Report
This manuscript provides a systematic review of the structural design of transparent polyimide (PI) films, focusing on the factors influencing their dielectric properties, water absorption properties, and optical transmittance, as well as the methods to synthesize high-performance PIs for high-speed communication. I have the following suggestions for improving the manuscript:
1. In the manuscript, the author states that PIs are replacing liquid crystal polymers (LCPs) as insulating materials due to being cost-effective and easy to process. However, the challenges of LCPs as insulating materials and the advantages of PIs over LCPs are not mentioned. It is important to provide a brief description or comparison of these materials to support the author's statement.
2. The writing style of the article can be improved to enhance readability and compactness. Some paragraphs can be combined or trimmed to avoid repetition and improve the flow of the content. Specifically:
(a) The paragraph on page 2, line 48 and the part on page 6, line 280 appear to have unclear expressions and logic. These sections should be modified for clarity.
(b) The paragraphs "3. PI materials for 5G/6G applications" and "5. Material properties that need to be considered for 5G/6G applications" can be merged into one paragraph.
(c) The subtitle "4. Polyimide synthesis, characterization, and properties" covers three aspects, but only the synthesis process is discussed. Consider merging this subtitle into "6. Examples of structural designs of transparent PI films with low dielectric and low water‐absorbing properties."
3. It is suggested to provide a clear classification or framework of transparent polyimide films in the introduction section. This will help readers understand the different types and their characteristics, establishing a foundation for later discussions. Especially, why is the transparent PI material so important for 5G/6G communication? Additionally, in the discussion section, the author should compare the advantages and disadvantages of the various types of transparent polyimide films in terms of dielectric properties, water absorption properties, and optical transmittance. The inclusion of quantitative or qualitative evaluation criteria would further enhance the understanding of the differences between these films.
4. The conclusion statement "The inclusion of side groups can affect the properties of a PI both positively and negatively" is too general. The author should provide further elaboration to explain how side groups specifically impact the properties of polyimides.
5. In the conclusion section, where the main problems and challenges faced by polyimide films in communication applications are discussed, it would be beneficial to include experimental data or case studies to support the author's claims. Instead of solely describing theoretical possibilities and difficulties, the inclusion of concrete and feasible solutions or suggestions based on current research and experimental data would make the review more comprehensive.
The English proficiency of the manuscript is generally good. The language used is clear and understandable, allowing for effective communication of ideas. However, there are some areas where improvements can be made to enhance the overall quality of the English writing.
1. Clarity and Conciseness: While the manuscript effectively conveys information, there are areas where the writing can be more concise and focused. Reducing unnecessary repetition, eliminating redundant phrases, and streamlining the overall content would enhance the clarity and impact of the manuscript.
2. Organization and Structure: The manuscript generally follows a logical structure, with sections and subheadings that aid in navigating the content. However, there are instances where paragraphs or sections could be rearranged or combined to improve the flow and coherence of the information presented. Ensuring a smooth transition between ideas and sections would further enhance the overall organization.
Author Response
REVIEWER #1
General: This manuscript provides a systematic review of the structural design of transparent polyimide (PI) films, focusing on the factors influencing their dielectric properties, water absorption properties, and optical transmittance, as well as the methods to synthesize high-performance PIs for high-speed communication. I have the following suggestions for improving the manuscript:
Response: We are very grateful to the Reviewer’s various valuable comments. We carefully revised our manuscript based on the reviewer’s comments. Here are the responses to the comments raised by the Reviewer.
- In the manuscript, the author states that PIs are replacing liquid crystal polymers (LCPs) as insulating materials due to being cost-effective and easy to process. However, the challenges of LCPs as insulating materials and the advantages of PIs over LCPs are not mentioned. It is important to provide a brief description or comparison of these materials to support the author's statement.
Response: Authors are thankful to the reviewer for the helpful suggestion, and it has been carefully implemented in the Section 2 : L91-103. Further, we added “However, these materials suffer … microstructure of a ceramic” Page 2, L85-88. For this revision, we added three new references [19-21].
- The writing style of the article can be improved to enhance readability and compactness. Some paragraphs can be combined or trimmed to avoid repetition and improve the flow of the content. Specifically:
(a) The paragraph on page 2, line 48 and the part on page 6, line 280 appear to have unclear expressions and logic. These sections should be modified for clarity.
Response: Thanks to the reviewer for the suggestions and the manuscript has been modified accordingly. section 1: Page 2, line 48 and section 3.2.6: page 5, line 223-225.
(b) The paragraphs "3. PI materials for 5G/6G applications" and "5. Material properties that need to be considered for 5G/6G applications" can be merged into one paragraph.
Response: Thank you for your comments. The manuscript has been modified with suitable rearrangement to improve the readability according to your kind suggestion. In the revised manuscript, we have re-edited as follows;
Section 2: Materials with low Dk and low Df for 5G/6G applications
Section 3: PIs for 5G/6G applications
Section 3.1. General properties of PIs in terms of 5G/6G applications
Section 3.2. Properties of PIs that need to be considered for 5G/6G
applications
Section 4: Structural designs of transparent PI films with low dielectric and low
water-absorbing properties
Section 4.1. Polyimide synthesis, characterization, and properties
Section 4.2. Examples of structural designs of transparent films with low dielectric and low water-absorbing properties
We highlighted the re-organized sections in green in this revised manuscript, while we marked revised phrases and sentences in blue.
(c) The subtitle "4. Polyimide synthesis, characterization, and properties" covers three aspects, but only the synthesis process is discussed. Consider merging this subtitle into "6. Examples of structural designs of transparent PI films with low dielectric and low water‐absorbing properties."
Response: We thank the reviewer for pointing the connectivity. Now manuscript has been revised as above according to the reviewer’s suggestion. In the revised manuscript, we re-edited as follows;
Section 4: Structural designs of transparent PI films with low dielectric and low water-absorbing properties
Section 4.1. Polyimide synthesis, characterization, and properties
Section 4.2. Examples of structural designs of transparent films with low
dielectric and low water-absorbing properties
We highlighted the re-organized sections in green in this revised manuscript, while we marked revised phrases and sentences in blue.
- It is suggested to provide a clear classification or framework of transparent polyimide films in the introduction section. This will help readers understand the different types and their characteristics, establishing a foundation for later discussions. Especially, why is the transparent PI material so important for 5G/6G communication? Additionally, in the discussion section, the author should compare the advantages and disadvantages of the various types of transparent polyimide films in terms of dielectric properties, water absorption properties, and optical transmittance. The inclusion of quantitative or qualitative evaluation criteria would further enhance the understanding of the differences between these films.
Response: Thank you for your comments. According to your kind advice, our manuscript has been revised by including the classification of CPIs, the importance of CPIs in 5G/6G communication applications, and other things (For this revision, we newly added Figure 2, and corresponding discussion with two new references ([55 and 60]) in the Section 3.2.6: page 6, L277-page 7, L296. However, we already included the dielectric properties, water absorption, and optical transmittance data in the Table 1, for all the articles reviewed in the manuscript. Further we added “Dimensional stability … low thermal expansion. [60].” To the section 3.2.7 from L329-334.
- The conclusion statement "The inclusion of side groups can affect the properties of a PI both positively and negatively" is too general. The author should provide further elaboration to explain how side groups specifically impact the properties of polyimides.
Response: Thank you for your kind advice. The statement has been elaborated now as per the suggestion (Page 29, L791-799; L809-812).
- In the conclusion section, where the main problems and challenges faced by polyimide films in communication applications are discussed, it would be beneficial to include experimental data or case studies to support the author's claims. Instead of solely describing theoretical possibilities and difficulties, the inclusion of concrete and feasible solutions or suggestions based on current research and experimental data would make the review more comprehensive.
Response: We are thankful to the reviewer for the comment and the manuscript has been modified accordingly. For instance, we discussed the experimental data related to the inclusion of side groups to the main chain and substitution of F and CF3 groups on the change of dielectric properties (Page 29, L791-799; L809-812). In addition, we added one more line to discuss about the challenges on PIs towards the communication applications (Section 6: page 29, L831-8432) “such as to solve low adhesiveness, cracks at the passivation layer, and etc. during soldering, cutting, and punching.”
Comments on the Quality of English Language
The English proficiency of the manuscript is generally good. The language used is clear and understandable, allowing for effective communication of ideas. However, there are some areas where improvements can be made to enhance the overall quality of the English writing.
- Clarity and Conciseness: While the manuscript effectively conveys information, there are areas where the writing can be more concise and focused. Reducing unnecessary repetition, eliminating redundant phrases, and streamlining the overall content would enhance the clarity and impact of the manuscript.
- Organization and Structure: The manuscript generally follows a logical structure, with sections and subheadings that aid in navigating the content. However, there are instances where paragraphs or sections could be rearranged or combined to improve the flow and coherence of the information presented. Ensuring a smooth transition between ideas and sections would further enhance the overall organization.
Response: We are deeply thankful to the reviewer for the comments. According to your kind and valuable advices, we have re-organized the Sections from 2 to 4, as we replied to your comments as above. In this way, we believe that the clarity and conciseness as well as the organization and structure had been significantly improved.
We highlighted the re-organized sections in green in this revised manuscript, while we marked revised phrases and sentences in blue.

Reviewer 2 Report
This a very good review that could be published in Nanomaterials. One optional suggestion is to include a more extended introduction that showcases the integration structure, as it would be very beneficial for the readers.
Additional small technical remarks can be seen in the attached file,

Author Response
REVIEWER #2
General This a very good review that could be published in Nanomaterials. One optional suggestion is to include a more extended introduction that showcases the integration structure, as it would be very beneficial for the readers.
Response: We are deeply thankful to the Reviewer for kind suggestions towards the improvement of our manuscript and we made all corrections according to the Reviewer and the suggested modifications were kept in blue color. We added a few more sentences in the Introduction (Page 2, Lines 47-51). In addition, we re-organized the Sections 2 to 4 to help more clear and concise readability. We hope that this revised manuscript may convey more effectively necessary information for readers.
In addition, we slightly modified a few phrases for enhancing the clarity of the meanings in the manuscript as follows;
Lines 10: modified as “low dielectric constant (Dk) (<2.5) materials.”
Lines 52: modified as “low dielectric constant.”
Line 821: modified as “low-Dk PI-based materials.”
Line 869: now deleted “4” in the reference 4.
We highlighted the re-organized sections in green in this revised manuscript, while we marked revised phrases and sentences in blue.
